Resource

# PRMT1 is required for the generation of MHC-associated microglia and remyelination in the central nervous system

Jeesan Lee[1], Oscar David Villarreal[1], Yu Chang Wang[2], Jiannis Ragoussis[2], Serge Rivest[3], David Gosselin[3], Stéphane Richard[1]

Remyelination failure in multiple sclerosis leads to progressive demyelination and inflammation, resulting in neurodegeneration and clinical decline. Microglia are innate immune cells that can acquire a regenerative phenotype to promote remyelination, yet little is known about the regulators controlling the regenerative microglia activation. Herein, using a cuprizone (CPZ)-diet induced de- and remyelination mice model, we identify PRMT1 as a driver for MHC-associated microglia population required for remyelination in the central nervous system. The loss of PRMT1, but not PRMT5, in microglia resulted in impairment of the remyelination with a reduction of oligoprogenitor cell number and prolonged microgliosis and astrogliosis. Using single-cell RNA sequencing, we found eight distinct microglial clusters during the CPZ diet, and PRMT1 depleted microglia hindered the formation of the MHC-associated cluster, expressing MHCII and CD11c. Mechanistically, PRMT1-KO microglia displayed reduced the H3K27ac peaks at the promoter regions of the MHC- and IFN-associated genes and further suppressed gene expression during CPZ diet. Overall, our findings demonstrate that PRMT1 is a critical regulator of the MHC- and IFN-associated microglia, necessary for central nervous system remyelination.

## Introduction

Microglia are tissue-resident innate immune cells of the central nervous system (CNS), constituting 5–12% of the CNS population (Wolf et al, 2017). Microglia patrol the CNS to recognize homeostatic disturbances, and quickly change their transcriptional program, morphology, and electrophysical properties to perform a wide range of roles (Wolf et al, 2017; Prinz et al, 2019). For instance, microglia use unique cell surface-receptors to sense pathogens, apoptotic cells, and protein aggregates, and then rapidly switch to a phagocytic phenotype to eliminate these materials (Rivest, 2009; Hammond et al, 2021). Furthermore, upon sensing any injuries to the CNS, microglia proliferate, migrate to the site of the damage, and secrete cytokines, chemokines, nitric oxide, and reactive oxygen species to modulate the injury-induced immune response (Li & Barres, 2018). Increasing studies using single-cell sequencing technology have unveiled the dynamic phenotypes of microglia during development and pathological conditions (Masuda et al, 2020). However, the molecular mechanisms or signaling pathways leading to expansion of specific microglial subpopulations during CNS pathophysiology remains underexplored.

Cuprizone is a copper chelating agent that results in demyelination, a pathological hallmark of multiple sclerosis (Vega-Riquer et al, 2019). Administration of a 0.2% cuprizone diet (CPZ) in mice induces apoptosis of oligodendrocytes, activating microglia and astrocytes (Lloyd & Miron, 2019; Vega-Riquer et al, 2019). Under these conditions, microglia recognize damaged oligodendrocytes and become the main phagocytic cell type to clear myelin debris, which allows the recruitment of oligoprogenitor cells (OPCs) at the sites of demyelination (Lloyd & Miron, 2019). In addition, microglia secrete various pro-regenerative factors to aid in OPC proliferation, migration, and differentiation (Lloyd & Miron, 2019; Mahmood & Miron, 2022). Emerging evidence implies that a transcriptionally distinct, CD11c[+] subset of microglia could promote the remyelination process (Wlodarczyk et al, 2017); however, the molecular mechanisms regulating this subset are poorly understood (Lloyd & Miron, 2019; Masuda et al, 2020).

Post-translation modifications are instrumental in modifying protein stability, interactions, and localization, and thereby impact a diverse array of signaling pathways (Xu & Richard, 2021). Among these post-translation modifications, arginine methylation is catalyzed by protein arginine methyltransferases (PRMTs), which use S-adenosyl-L-methionine (AdoMet) to transfer a methyl groups to arginine (Bedford & Clarke, 2009). PRMTs are categorized according

[1]Segal Cancer Center, Lady Davis Institute for Medical Research and Gerald Bronfman Department of Oncology and Departments of Biochemistry, Human Genetics, and Medicine, McGill University, Montreal, Canada   [2]McGill Genome Centre, Department of Human Genetics, McGill University, Montreal, Canada   [3]Neuroscience Laboratory, CHU de Quebec Research Center and Department of Molecular Medicine, Faculty of Medicine, Laval University, Quebec City, Canada

Correspondence: stephane.richard@mcgill.ca; david.gosselin@crchudequebec.ulaval.ca

to the types of methylation they catalyze (Bedford & Clarke, 2009). PRMT1 is the major type I enzyme responsible for catalyzing monomethylarginine (MMA) and asymmetric dimethylarginine (aDMA), whereas PRMT5 is the major type II enzyme and generating MMA and symmetric dimethylarginine (sDMA) (Rotshenker, 2009). The substrates of PRMTs include RNA binding proteins, DNA damage repair proteins, transcription factors, signaling proteins, and histones (Guccione & Richard, 2019; Xu & Richard, 2021). In recent years, PRMTs have been shown to play instrumental roles in immune cell differentiation, activation, and viral-mediated type I IFN responses by methylating proteins and histones (Blanc & Richard, 2017). Nevertheless, the role of PRMTs in modulating microglia function during de/remyelination processes has never been investigated.

In the present article, we report that PRMT1 is a regulator of the MHC- and IFN-associated microglia cluster using the CPZ mouse model. Mice with microglia deficient for PRMT1 using the $Cx3cr1^{CreERT}$ driver were generated ($PRMT1^{FL/FL;Cx3cr1-CreERT}$). Lack of the MHC-associated microglia cluster in tamoxifen (TAM) treated $PRMT1^{FL/FL;Cx3cr1-CreERT}$ mice correlates with a failure to induce CNS remyelination after demyelination induced by CPZ.

# Result

## CNS remyelination defects and microgliosis in cuprizone induced demyelination in $PRMT1^{Cx3cr1-KO}$ mice

To investigate the role of arginine methylation in microglia, we generated mice with microglia deficient for PRMT1 or PRMT5 using the $Cx3cr1^{CreERT}$ driver (Fig S1A). Microglia isolated from tamoxifen (TAM) injected $PRMT1^{FL/FL;Cx3cr1-CreERT}$ ($PRMT1^{Cx3cr1-KO}$) and $PRMT5^{FL/FL;Cx3cr1-CreERT}$ ($PRMT5^{Cx3cr1-KO}$) exhibited the loss PRMT1 and PRMT5 with reduction in the histone marks, H4R3me2a and H4R3me2s, respectively (Fig S1A). The impact of PRMT1 and PRMT5 ablation on microglia number and morphology was assessed using anti–Iba-1 antibody. During the resting state, $PRMT1^{Cx3cr1-KO}$ or $PRMT5^{Cx3cr1-KO}$ microglia did not show any differences in the cell number in the corpus callosum (CC), cortex, and hippocampus of the CNS compared with their respective controls, $PRMT1^{FL}$ and $PRMT5^{FL}$ (Figs 1A and S1B). However, distinct morphological changes were observed only in the $PRMT1^{Cx3cr1-KO}$ microglia (Fig 1B) and not in $PRMT5^{Cx3cr1-KO}$ microglia (Fig S1C). The IMARIS based detail morphological analysis showed that $PRMT1^{Cx3cr1-KO}$ microglia had significantly longer total dendritic length with increased number of branches, segments, and terminals compared with control $PRMT1^{FL}$ (Fig 1B).

Because PRMT1 and PRMT5 were shown to be essential in oligodendrocyte precursor cells for their differentiation and CNS myelin in mice (Hashimoto et al, 2016; Scaglione et al, 2018), we asked whether microglia PRMT1 or PRMT5 could influence CNS demyelination and remyelination using the cuprizone (CPZ) mouse model of myelin lesion (Gudi et al, 2009). $PRMT1^{Cx3cr1-KO}$ and $PRMT5^{Cx3cr1-KO}$ mice were fed 0.2% CPZ diet for 5 wk to instigate CNS demyelination followed by 1 wk of normal diet to induce remyelination (Gudi et al, 2014). We focused on the CC area to qualitatively visualize the demyelination and remyelination. At 5 wk of CPZ diet, myelin was completely absent in the CC, and there were no significant differences in the extent of demyelination between the

genotypes (Figs 1C and D and S1D and E). However, upon changing to a normal diet for 1-wk, significant remyelination was visible in the $PRMT1^{FL}$, but not in the $PRMT1^{Cx3cr1-KO}$ mice (Fig 1C and D). Conversely, the loss of PRMT5 in microglia did not impair the CNS remyelination (Fig S1D and E). Microglia accumulate at the CC during the demyelination period and dissipate after the remyelination (Gudi et al, 2014). Consistent with this, microglia significantly increased during the CPZ diet (microgliosis; Iba1+) and subsequently decreased upon changing to a normal chow (+1 wk off) in CC of $PRMT1^{FL}$ mice (Fig 1E and F) and $PRMT5^{FL}$ and $PRMT5^{Cx3cr1-KO}$ mice (Fig S1F and G). In contrast, in the $PRMT1^{Cx3cr1-KO}$ mice, microglia (Iba1+) significantly increased during the CPZ diet and remained in the CC upon changing to a normal chow (+1 wk off) (Fig 1E and F). Thus, $PRMT1^{Cx3cr1-KO}$ mice have defects in remyelination with prolonged microgliosis in the CPZ demyelination mouse model.

## $PRMT1^{Cx3cr1-KO}$ mice have prolonged gliosis and reduced number of oligoprogenitor cells during the remyelination phase

One possible explanation for remyelination failure observed in the $PRMT1^{Cx3cr1-KO}$ mice was the inability of microglia to clear the myelin debris during the demyelination phase. Thus, we performed transmission electron microscope (TEM) analysis and measured the myelin lamellae and myelin debris during the normal and CPZ diet 5 wk in cross-sections of the CC of $PRMT1^{FL}$ and $PRMT1^{Cx3cr1-KO}$ mice (Fig 1G and H). The TEM analysis showed a complete loss of the myelin layer during the CPZ diet in both $PRMT1^{FL}$ and $PRMT1^{Cx3cr1-KO}$ mice and no myelin debris was observed in either genotype, suggesting that the phagocytosis activity of PRMT1-deficient microglia was not impaired.

The remyelination process is normally preceded by the proliferation of oligoprogenitor cells (OPCs), which subsequently differentiate into myelinating oligodendrocytes (Matsushima & Morell, 2001). Therefore, we first analyzed the number of OPCs by staining with anti–Olig2-antibody (Yokoo et al, 2004). In $PRMT1^{FL}$ mice, Olig2+ cells increased in the CC after changing to a normal diet (+1 wk off), but this increase was impaired in the $PRMT1^{Cx3cr1-KO}$ mice (Fig 1I and J).

We next asked whether extension of the normal chow for 3 wk (+3 wk off) would recover the remyelination defects observed in $PRMT1^{Cx3cr1-KO}$ mice. We did not recover the remyelination phenotype, as we still observed severe impairment of remyelination, reduction in the recruitment of Olig2+ cells at the site of demyelination, accumulation of Iba1+ microglia and GFAP+ astrocytes in $PRMT1^{Cx3cr1-KO}$ CC compared with the $PRMT1^{FL}$ (Fig S2A–D). To see whether these accumulated microglia were activated, we stained for Mac2, a marker of highly phagocytic microglia (Rotshenker, 2009). The expression of Mac2+ cells was significantly increased in the $PRMT1^{Cx3cr1-KO}$ mice compared with the $PRMT1^{FL}$ control (Fig S2E), suggesting the microglia from $PRMT1^{Cx3cr1-KO}$ mice are continuously activated during the remyelination phase with increased gliosis and decreased OPC recruitment.

## Single-cell RNA-sequencing revealed a lack of major histocompatibility complex (MHC)–associated microglia cluster in $PRMT1^{Cx3cr1-KO}$ mice

To closely inspect the microglia heterogeneity during the CPZ diet in $PRMT1^{FL}$ and $PRMT1^{Cx3cr1-KO}$ mice, we performed single-cell RNA

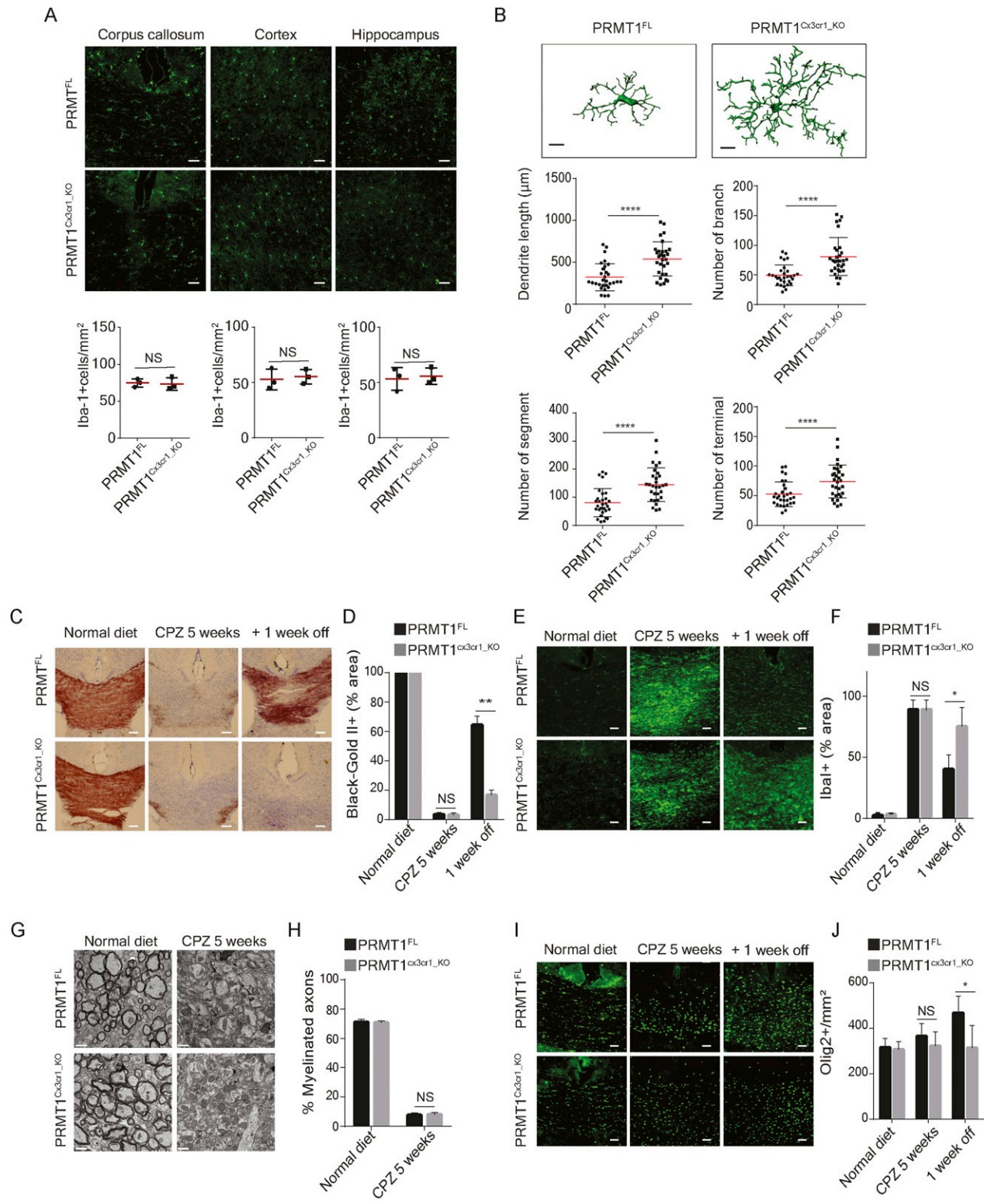

**Figure 1. Central nervous system (CNS) remyelination defects in *PRMT1^Cx3cr1-KO* mice using cuprizone mouse model of demyelination.**
**(A)** Iba-1(green) staining in the cortex, hippocampus and corpus callosum (CC) of *PRMT1^FL* (n = 3), and *PRMT1^Cx3cr1-KO* (n = 3) CNS. Representative images are shown. Scale bars represent 50 μm. Data presented as mean ± SEM. *P*-values were analyzed using Mann–Whitney U-statistical test (NS, not significant). **(B)** IMARIS-based morphological analysis of microglia from *PRMT1^FL* (n = 3), and *PRMT1^Cx3cr1-KO* (n = 3) CNS. Representative images are shown. 10 cells were analyzed for dendritic length, number of branches, number of segments, and terminals and presented as a dot plot. Scale bars represent 10 μm. Data presented as mean ± SEM. *P*-values were analyzed using Mann–Whitney U-statistical test (**P* < 0.05, *****P* < 0.0001). **(C)** Mice fed with 0.2% cuprizone diet (CPZ) were euthanized according to the following time points: Normal diet,

sequencing (scRNA-seq) using the 10X Genomics platform. Mouse brains were enzymatically dissociated into single-cell suspension and sorted for CD45$^+$ and CD11b$^+$ to isolate microglia and macrophages. A total of 10,000 sorted cells at CPZ diet 5 wk were sequenced from two separate PRMT1$^{FL}$ and PRMT1$^{Cx3cr1-KO}$ mice. Using the Seurat software package, a total of 6,000 microglia per genotype passed the quality control. The Uniform Manifold Approximation and Projection (UMAP) analysis revealed nine transcriptionally defined clusters (labelled A to F) with unique gene sets in PRMT1$^{FL}$ and PRMT1$^{Cx3cr1-KO}$ microglia (Fig 2A and Table S1).

Eight clusters were microglia populations and there was a minor cluster (cluster F) representing 2% of the cells that expressed monocyte markers (Ccr2, Mrc1, Mgl2, and F13a1) (Figs 2B and C and S3). Thus, brain infiltrating monocytes were minimally present during the CPZ-diet induced demyelination and remyelination process.

Notably, single-cell mapping showed altered microglia composition in PRMT1$^{Cx3cr1-KO}$ microglia, with the loss of C1 (11%), C2 (3%), and D (3%) populations with gains of B1 (9%) and E (24%) populations (Fig 2A–C). Close inspection revealed that the B1 cluster expressed transcripts found in disease-associated microglia phenotype (Spp1, Cybb, Apoe, and Cd38) (Fig 2C). The cluster E expressed mixtures of homeostatic microglia signature (P2ry12 and Sall1) and genes associated with pro-inflammatory transcripts (Tnfaip2 and Nfkbia) (Fig 2C). The transition of microglia during the CPZ diet is poorly defined. Therefore, we subsequently used the scVelo algorithm, which uses the ratio of spliced to unspliced transcripts to infer RNA velocity and predict the transition of the clusters (Bergen et al, 2020), similar to previous work by our group (Couturier et al, 2020). In the PRMT1$^{FL}$, the scVelo analysis showed two divergent trajectories of microglia, A1 to A2 clusters and B2 to C1 to C2 to D clusters (Fig 3A). The cluster A1 exclusively expressed (Birc5, Mki67, Cdk1, and Ccnb2) which are indicative of proliferating cells in addition to oxidative phosphorylation (OXPHOS) associated genes (Figs 2C and S3). Cluster A2 also expressed OXPHOS genes (Rps26, Ndufa1, Cox6a2, and Uqcc2) and the cytoplasmic translation processing genes (Rps26, Rpl41, and Rps8) (Figs 2C and S3). Therefore, we will now call these A1 and A2 clusters "proliferating population," and these subsets possibly emerge from the stress and inflammatory condition exerted by the CPZ diet.

The B2 cluster was enriched for transcripts linked to phagolysosome (Ctsl, Ctsd, Cspg4, Gusb, and Ctsb) and lipid recycling (Lpl and Lrp1), signifying that the B2 cluster are phagocytic microglia likely involved in the demyelination from CPZ (Figs 2C and S3). B2 cluster transitioned to C1 to C2 clusters of which C1 and C2 express high levels of IFN-associated transcripts (Irf7, Oasl2, Stat1, and Itgax), pro-inflammatory cytokine and chemokine transcripts (Il1b, Il1a, Tnf, Cxcl10, and CCrl2), and MHCI and II processing transcripts (CD74, H2-Q4, H2-Ab1, and H2-D1) (Figs 2C and S3). Therefore, we will call the B2 cluster the "phagocytic population" and C1 to C2 clusters the "MHC-associated population." Finally, the MHC-associated cluster transitioned to the D cluster, which expressed homeostatic microglia markers (Tmem119, Seplg, Siglech, and Sox4) (Fig 2C). In stark contrast, the loss of PRMT1 completely blocked the transition from B2 to C1 cluster, thus abolishing the MHC-associated and homeostatic clusters (Fig 3A and B). To summarize, wild-type microglia adopt two defined trajectories upon CPZ insult, proliferating microglia or phagocytic/MHC-associated/homeostatic microglia, but the loss of PRMT1 impedes the transition from phagocytic to MHC-associated and ultimately to homeostatic microglia (Fig 3B).

We further performed monocle analysis, which aligns the cells in the pseudotime trajectory (Qiu et al, 2017) (Fig 3C). In a similar fashion as scVelo analysis, PRMT1$^{FL}$ microglia begin with the proliferative state (A1 and A2 clusters), which then transition to phagocytic (B2 cluster), MHC-associated clusters (C1 and C2), and homeostatic state (Fig 3D). In PRMT1$^{Cx3cr1-KO}$ microglia, proliferating and phagocytic microglia are dominant populations that are hindered to transition to the MHC-associated and homeostatic populations (Fig 3D and E). We further confirmed the single-cell analysis results by performing FACS on PRMT1$^{FL}$ and PRMT1$^{Cx3cr1-KO}$ microglia during the CPZ diet. We analyzed the CD11c+ and MHCII+ microglia for their presence of markers of the MHC-associated cluster. Consistently, during the CPZ diet, we found an increased number of MHCII+ (35.3%) and CD11c+ (37.7%) microglia in the PRMT1$^{FL}$ (Fig 4A–D). In contrast, PRMT1$^{Cx3cr1-KO}$ microglia did not increase the MHCII+(5.64%) and CD11c+ (9.15%) populations during the CPZ diet (Fig 4A–D). These data suggest that PRMT1$^{Cx3cr1-KO}$ mice are deficient in the MHC-associated microglia cluster expressing MHCII and CD11c during the CPZ diet.

## H3K27ac peaks of MHC-associated transcripts are lost in PRMT1-deficient microglia

PRMT1 methylates histone 4 arginine 3 (H4R3me2a) (Wang et al, 2001) activating transcription with the subsequent acetylation of histones (Huang et al, 2005). Genome-wide H4R3me2a are notoriously impossible to perform as enrichment is not observed for chromatin immunoprecipitation (ChIP) (Guccione & Richard, 2019). Combined with a relatively small yield of microglia cells per mouse, H4R3me2a ChIP-seq was thus not feasible. As an alternative, we performed H3K27ac ChIP-sequencing (Table S2) and bulk RNA-sequencing (Table S3) in PRMT1$^{FL}$ and PRMT1$^{Cx3cr1-KO}$ microglia

CPZ diet 5 wk (n = 5) and CPZ diet 5 wk + normal diet 1 wk (+1 wk off) (n = 5). Black gold II staining (brown) was performed to visualize myelin and quantified at the CC of PRMT1$^{FL}$ (n = 5), and PRMT1$^{Cx3cr1-KO}$ (n = 5) CNS. Representative images are shown. Scale bars represent 50 μm. **(D)** Bar graph depicts average myelin staining in the CC. Data presented as mean ± SEM. P-values were analyzed using Mann–Whitney U-statistical test (**P < 0.01, NS, Not significant). Scale bars represent 50 μm. **(E, F)** Anti-Iba-1 (green) antibody staining was performed during the normal diet, CPZ diet 5 wk, and CPZ diet 5 wk + normal diet 1 wk in PRMT1$^{FL}$ (n = 5), and PRMT1$^{Cx3cr1-KO}$ (n = 5). Representative images are shown. Scale bars represent 50 μm. **(F)** Iba-1 staining was quantified at the CC and illustrated as a bar graph. Data presented as mean ± SEM. P-values were analyzed using Mann–Whitney U-statistical test (*P < 0.05, NS, not significant). **(G, H)** Representative transmission electron microscope image of myelinated axons from PRMT1$^{FL}$ (n = 3), and PRMT1$^{Cx3cr1-KO}$ (n = 3) at the indicated time points. **(H)** Myelinated axons were quantified and displayed in a bar graph. Data presented as mean ± SEM. P-values were analyzed using Mann–Whitney U-statistical test (NS, not significant). **(I, J)** Anti-Olig2 (green) antibody staining was performed during the normal diet, CPZ diet 5 wk and CPZ diet 5 wk + normal diet 1 wk in PRMT1$^{FL}$ (n = 3), and PRMT1$^{Cx3cr1-KO}$ (n = 3) CNS. Representative images are shown. Scale bars represent 50 μm. **(J)** Olig2 staining was quantified at the CC and illustrated as a bar graph. Data presented as mean ± SEM. P-values were analyzed using Mann–Whitney U-statistical test (*P < 0.05, NS, Not significant).

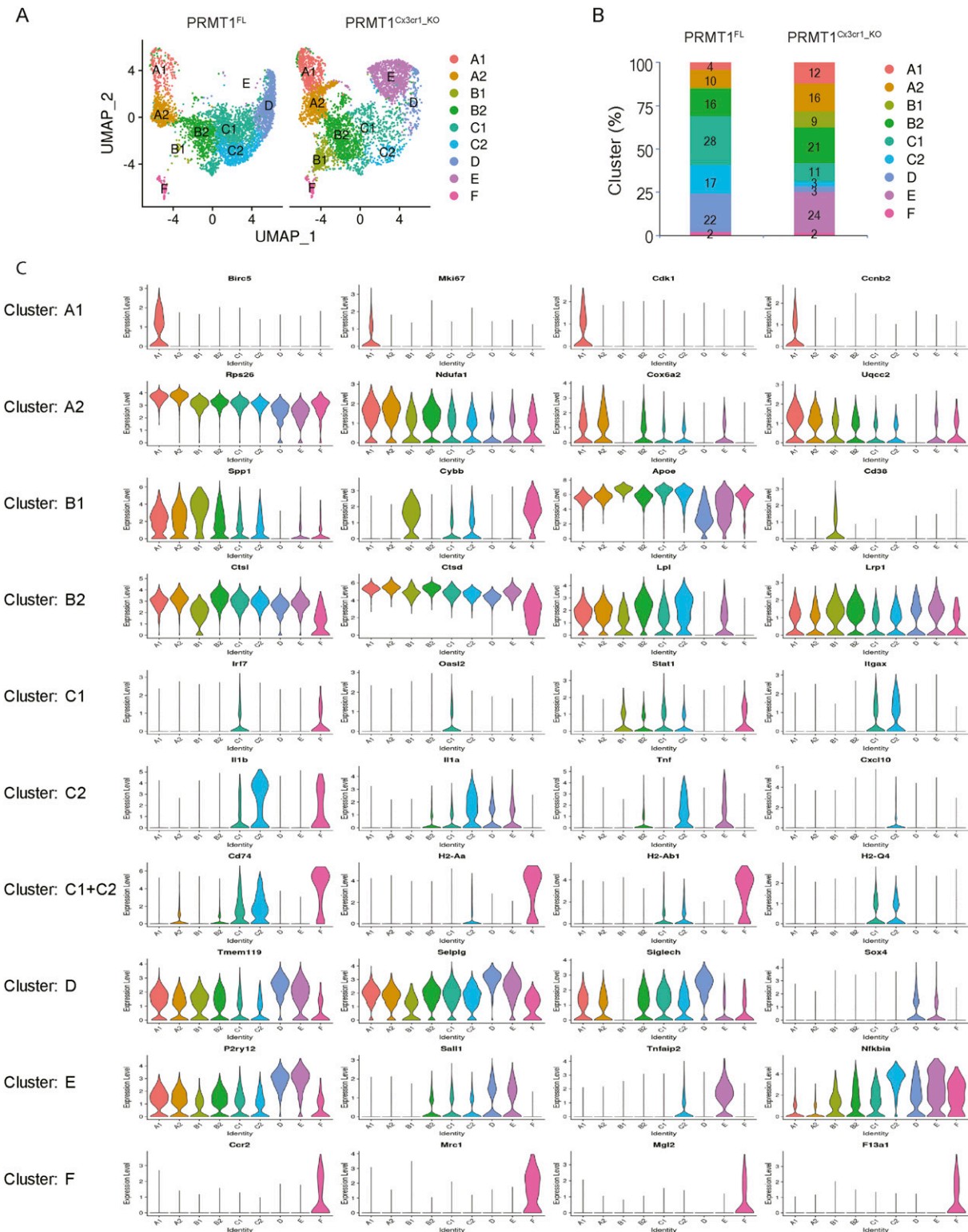

**Figure 2. Single-cell RNA sequencing analysis on *PRMT1^FL* and *PRMT1^Cx3cr1-KO* microglia from mice fed CPZ for 5 wk.**
**(A)** sc-RNA seq was performed from *PRMT1^FL*(n = 2) and *PRMT1^Cx3cr1-KO*(n = 2) mice during the CPZ diet 5 wk. Representative UMAP visualization of 6,000 microglia from each of the genotypes (n = 1) showing nine distinct clusters of microglia. **(B)** Stacked bar chart representing percentage of each cluster. **(C)** Violin plot showing marker genes for individual clusters.

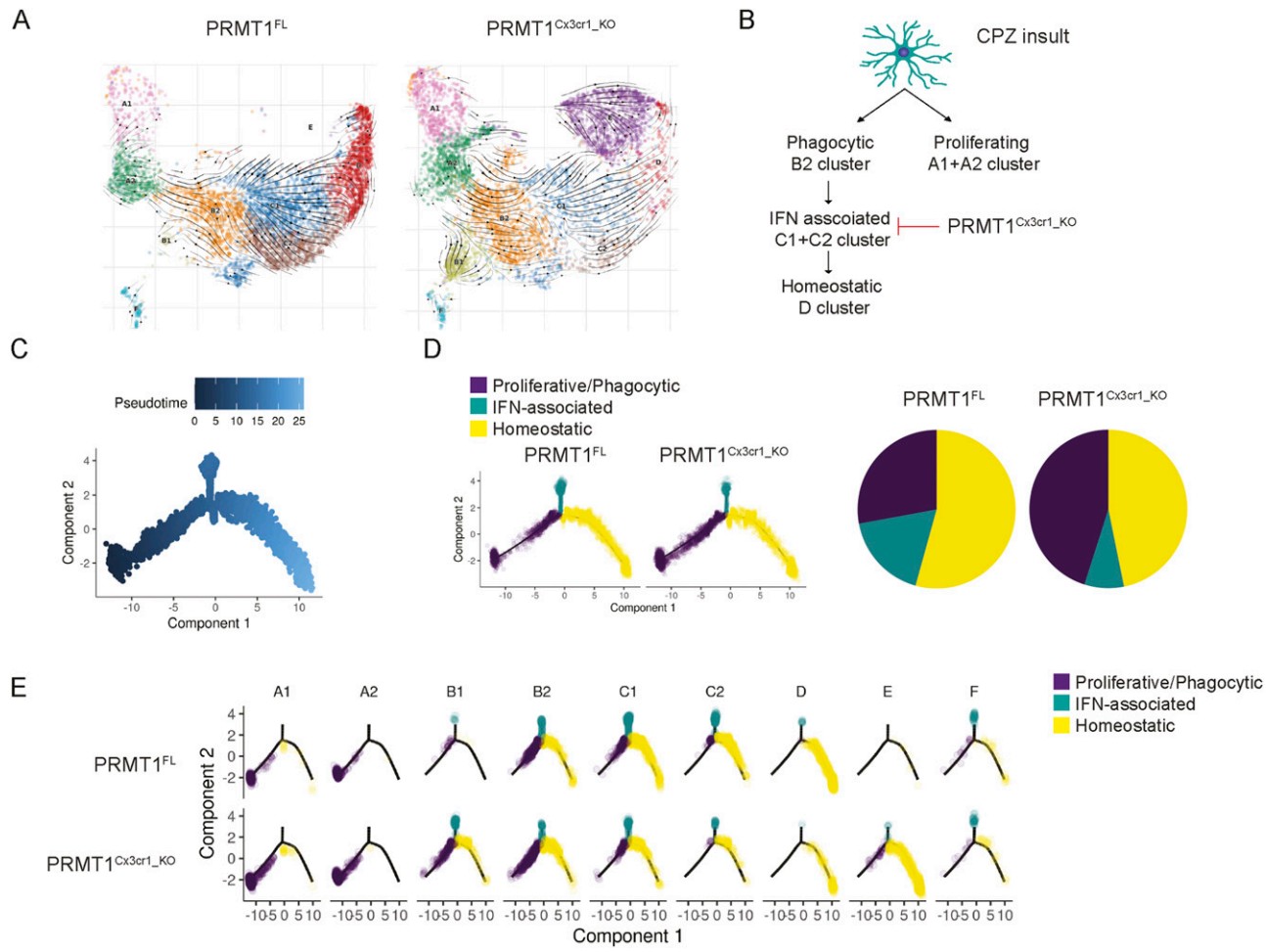

**Figure 3. The MHC-associated cluster is lost with *PRMT1* deficiency.**
**(A)** Trajectory of microglia was analyzed using scVelo algorithm and projected on the UMAP. Black arrowheads indicate the transition of microglia from one cluster to the other. **(B)** Schema illustrating the transition of clusters according to the scVelo analysis. **(C, D)** Monocle pseudotime trajectory analysis using the sc-RNA seq data from *PRMT1*^FL^ and *PRMT1*^Cx3cr1-KO^ microglia. **(C, D, E)** Cells are labelled according to (C), pseudotime, three defined states (D), and 9 clusters according to the three defined states (E).

during the normal and CPZ diet to decipher if PRMT1 could impact the transcriptional landscape of microglia (Fig 5A). Specifically, during the CPZ diet, we isolated *PRMT1*^FL^ microglia according to the CD11c+ and CD11c− populations (Fig 5A). The heat map and density plots show a striking increase of H3K27ac during the CPZ diet in both *PRMT1*^FL^ and *PRMT1*^Cx3cr1-KO^ microglia compared with the normal diet (Fig 5B and C). Interestingly, during the CPZ diet, the increase of H3K27ac peak and tag count densities was partially suppressed in the *PRMT1*^Cx3cr1-KO^ compared with the *PRMT1*^FL^. These findings suggest PRMT1 is required for the subsequent deposition of H3K27ac at specific promoters (Fig 5C), either by the deposition of H4R3me2a or the methylation of transcription factors or coactivators (e.g., histone acetyltransferases) that results in H3K27ac. Further analysis showed reduced H3K27ac peak at the gene promoter regions associated with IFN (*Ikkbe, Gadd24b, Irf1, Csf1,* and *Cd11c*) and MHCII (*H2-ab1, H2-eb1,* and *H2-aa*) in *PRMT1*^Cx3cr1-KO^ compared with *PRMT1*^FL^ (Fig 5D). Likewise, the decrease in the H3K27ac at the IFN and MHCII-associated genes reduced the total mRNA levels (Fig 5E). Collectively, our data point to a role of PRMT1

in transcriptionally activating MHC-associated genes for the generation of MHC-associated microglia clusters.

## Discussion

In the present article, we define a function for PRMT1 in microglia during CPZ-induced demyelination and remyelination in the CNS. We observed that PRMT1, but not PRMT5, was required to modify microglia to promote remyelination in the CNS. This failure in remyelination in PRMT1-deficient mice was also associated with prolonged microgliosis and astrogliosis as well as reduced OPCs recruited at sites of demyelination. Notably, PRMT1 regulated the generation of the MHC-associated microglia cluster known to express MHCII and CD11c during the entire demyelination and remyelination periods. The loss of PRMT1 reduced the gene expression of MHC-related genes in microglia by influencing the deposition of the activation mark, H3K27ac, at the promoters of

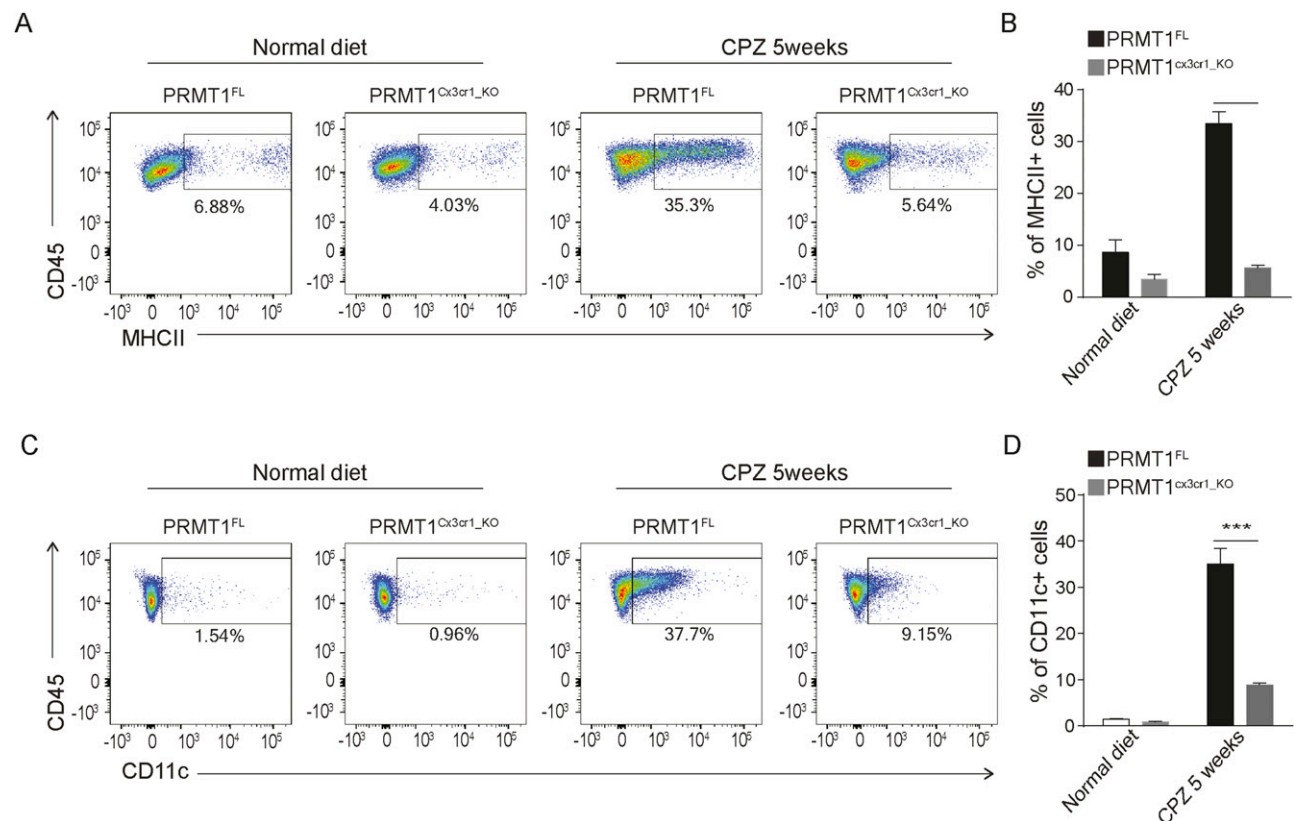

**Figure 4. The MHC-associated cluster expresses MHCII and CD11c.**
**(A)** Representative FACs analysis of MHCII+ microglia at the indicative time points in *PRMT1^FL^*(n = 3) and *PRMT1^Cx3cr1-KO^* (n = 3). **(B)** Bar graph visualizing the percent positive of MHCII microglia. Data presented as mean ± SEM. *P*-values were analyzed using Mann–Whitney U-statistical test (****P < 0.0001). **(C)** Representative FACs analysis of CD11c+ microglia during the normal and 5 wk of CPZ diet in *PRMT1^FL^*(n = 3), and *PRMT1^Cx3cr1-KO^* (n = 3). **(D)** Bar graph visualizing the percent of positive CD11c microglia. Data presented as mean ± SEM. *P*-values were analyzed using Mann–Whitney U-statistical test (***P < 0.001).

these genes and ultimately preventing their expression. Taken together, we identified a role for the PRMT1 in regulating the MHC-associated microglia population, indispensable for efficient remyelination in the CNS.

Our results show that H3K27ac marks are lost at the promoters of the MHC- and IFN-associated transcripts in PRMT1-deficient microglia. It is likely that the decrease in H3K27ac is the result of decreased presence of PRMT1-mediated H4R3me2a (Wang et al, 2001) at the promoters of these transcripts. Indeed, H4R3me2a is known to induce the subsequent acetylation of histones consistent with PRMT1 being essential for the establishment of active chromatin modifications (Huang et al, 2005). Although we propose a positive role for PRMT1 in MHCII gene expression in microglia, an opposite role for PRMT1, namely, as a repressor of MHCII transcription, has been reported in macrophages by promoting the arginine methylation and the degradation of the class II trans-activator (CIITA) (Fan et al, 2017). In macrophages, PRMT1 is needed for their differentiation into a more anti-inflammatory phenotype via H4R3me2a methylation at the PPARγ promoter (Tikhanovich et al, 2017). Moreover, methylation of TBK1 by PRMT1 is needed for its oligomerization and stimulation of type I IFN production (Yan et al, 2021). As a result of this, myeloid-specific PRMT1 knockout mice are more susceptible to viral infection (Tikhanovich et al, 2017; Yan et al, 2021).

The loss of PRMT5 in microglia did not have significant ramifications in the CNS during the cuprizone diet. We were surprised by these findings because PRMT5 regulates a diverse spectrum of lymphocyte biology and is involved in pattern recognition receptors sensing in macrophage (Cui et al, 2020; Ma et al, 2021). Furthermore, it is also required in oligodendrocyte progenitor cells for their differentiation and survival for CNS myelination by epigenetically regulating genes required for differentiation (Scaglione et al, 2018) and myelin basic protein (Branscombe et al, 2001). Because our study was limited to the cuprizone-diet model, we cannot rule out that PRMT5 might play a role during microglia development or in a different pathological context of the CNS.

We did not observe any significant change in the gene expression between microglia from wild type and *PRMT1^Cx3cr1-KO^* mice; however, we did observe a slight decrease in MHCII expression in the PRMT1-deficient microglia on a normal diet (Fig 4A and B), suggesting PRMT1 may affect the homeostatic function of microglia. However, upon CPZ-diet, the microglia loss of PRMT1 had a more profound impact on the regulation of microglia subpopulations derived from this brain injury. These findings suggest that PRMT1 is likely required for the proper activation of microglia to transition to specific microglia subsets such as the MHC-associated cluster.

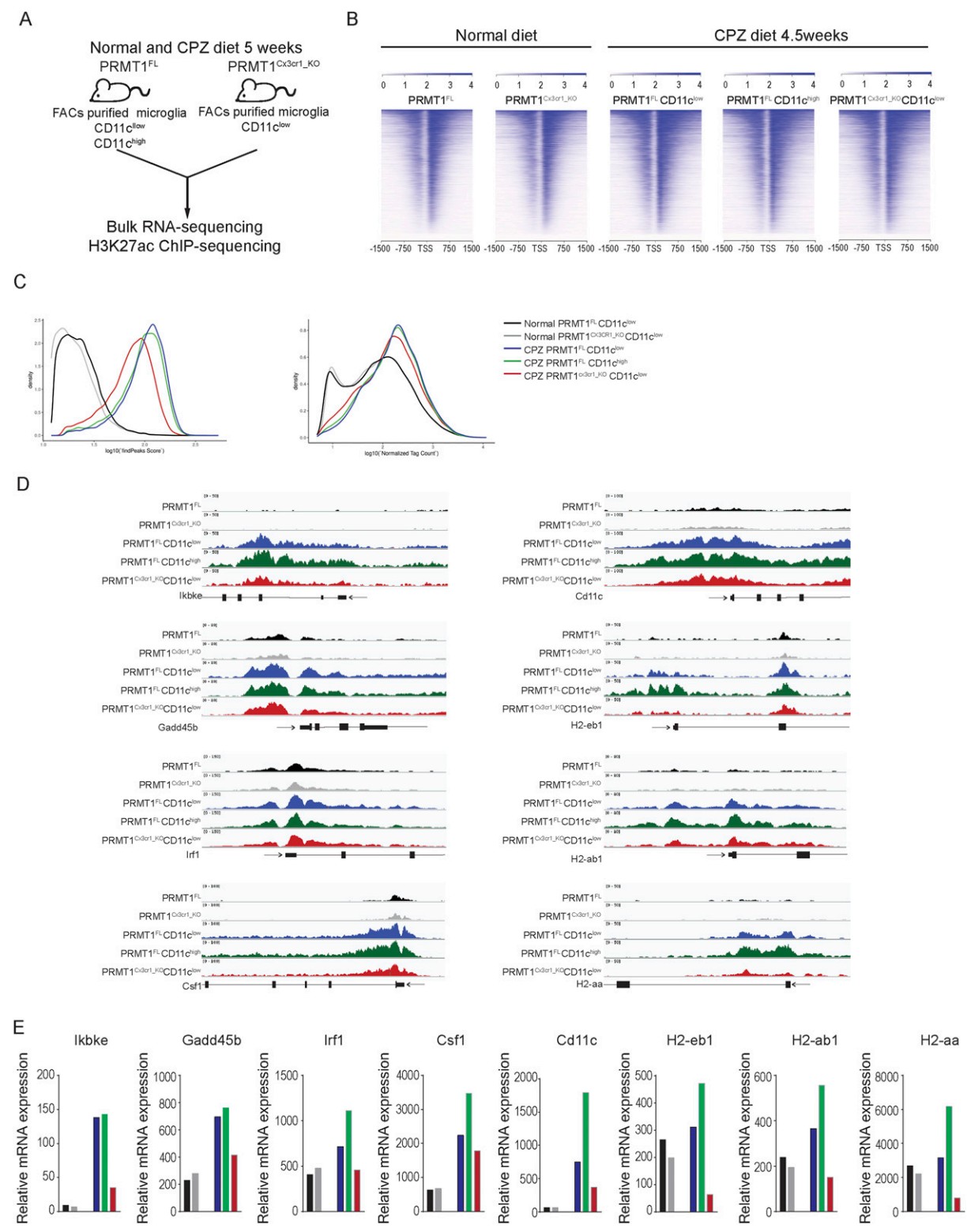

**Figure 5. PRMT1 is required for the H3K27ac deposition at the promoters of the MHC-associated genes.**
**(A)** Schematic illustrating ChIP-sequencing and RNA-sequencing of PRMT1$^{FL}$ and PRMT1$^{Cx3cr1-KO}$ microglia on a normal diet and CPZ diet for 5 wk. **(B)** H3K27ac enrichment at the transcriptional site (TSS) of genes in PRMT1$^{FL}$, PRMT1$^{Cx3cr1-KO}$, PRMT1$^{FL}$CD11c$^{low}$, PRMT1$^{FL}$CD11c$^{high}$, and PRMT1$^{Cx3cr1-KO}$CD11c$^{low}$ microglia. **(C)** H3K27ac peaks (left) and tag counts (right) were illustrated according to the densities. **(D)** A genome track of the IFN-associated loci including Ikbke, Gadd45b, Irf1, Csf1, Cd11c, H2-eb1, H2-ab1, and H2-aa displaying H3K27ac ChIP-seq in PRMT1$^{FL}$(black), PRMT1$^{Cx3cr1-KO}$(grey), PRMT1$^{FL}$CD11c$^{low}$(blue), PRMT1$^{FL}$CD11c$^{high}$(green), and PRMT1$^{Cx3cr1-KO}$CD11c$^{low}$(red) microglia. **(E)**mRNA-expression of MHC-associated genes in PRMT1$^{FL}$(black), PRMT1$^{Cx3cr1-KO}$(grey), PRMT1$^{FL}$CD11c$^{low}$(blue), PRMT1$^{FL}$CD11c$^{high}$(green), and PRMT1$^{Cx3cr1-KO}$CD11c$^{low}$(red) microglia.

Transcriptional changes in microglia during de/remyelination processes have been previously defined (Lloyd & Miron, 2019). Using scVelo and Monocle analysis to define the trajectory of microglia during demyelination and remyelination, we observed that microglia begin as a proliferative population (*Ki67* and *Cdk1*), transitioning to phagocytic (*Ctsl* and *Lpl*), IFN-associated (*Irf7* and *Oasl2*), and MHC-associated (*Cd74* and *H2-Aa*) subpopulations before becoming homeostatic (*Tmem119* and *Seplg*). The proposed trajectory is reminiscent of the transitional states of phagocytic APCs found in the periphery such as dendritic cells and macrophages (Kambayashi & Laufer, 2014).

We did not add a transcriptional inhibitor (e.g., flavopiridol or actinomycin D) to our buffers during the microglial isolation procedure. We did not find evidence that expression of CCL3, CCL4, JUN, and HSPA1A mRNAs, which are markers of spurious transcriptional induction associated with enzymatic-based isolation of microglia (Marsh et al, 2022) drove a specific cluster based on single-cell RNA-seq data. For example, CCL3 and CCL4 were elevated in clusters B2 and C2, JUN in D, and HSPA1A in F (Table S1). Given this, genotypes and experimental conditions, but not the isolation procedure used, were likely the main variables driving the clusters identified in our analyses.

The exact cellular functions of the MHC-associated cluster enabled by PRMT1 that promote remyelination of the CNS remain to be elucidated. For example, does MHC-associated microglia directly impact remyelination by secreting neurotrophic factors to promote OPCs proliferation or differentiation? During postnatal development or de/remyelination pathology, the CD11c+ microglia population expands at the white matter regions and secrete neurotrophic factor such as Igf1 to promote myelinogenesis (Wlodarczyk et al, 2017). Analysis of our scRNA-seq data revealed that the MHC-associated cluster arose with de/remyelination pathology and exclusively expressed CD11c. *Igf1* was expressed in all clusters, except cluster D, with highest expression being in clusters A1, A2, B1, and B2 that do not change much in *PRMT1^{Cx3cr1-KO}* mice (Fig S4A). Although a loss of CD11c+ microglia population, as defined by protein expression, in *PRMT1^{Cx3cr1-KO}* microglia was observed, no significant decrease *Igf1* mRNA expression level was noted that could account for the remyelination phenotype we observe. *Igf1* expression increased significantly with CPZ diet in both WT_CD11cLo and KO_CD11cLo. The increase in *Igf1* mRNA of the KO_CD11cLo was similar to the WT_CD11cHi microglia (Fig S4B). Therefore, absence of PRMT1 does not compromise production of Igf1 by microglia which suggests that alternative defective mechanisms limit remyelination. Although we did not observe at 5 wk by TEM microscopy obvious defects in myelin debris clearance, we cannot rule out this mechanism completely as failure to do so hinders remyelination in the CPZ model (Lampron et al, 2015). Another possible explanation of remyelination failure in *PRMT1^{Cx3cr1-KO}* mice could be attributed to persistent microgliosis during the remyelination period. It has been shown that microglia undergo necroptosis to give rise to the pro-regenerative microglia population to promote CNS remyelination (Lloyd et al, 2019). Therefore, the MHC-associated cluster might play a significant role in eventually allowing for timely microglia cell death required for OPC migration and proliferation at demyelinating sites (Kirby et al, 2006; Hughes et al, 2013).

In conclusion, we report PRMT1 as a molecular driver of the MHC-associated microglia cluster during CNS demyelination. Therefore, strategies to enhance PRMT1 expression might be therapeutically beneficial for the promotion of CNS remyelination in demyelinated areas.

# Materials and Methods

### Mice

The *PRMT1^{FL}* (Yu et al, 2009) and *PRMT5^{FL}* (Calabretta et al, 2018) alleles were generated previously and maintained on the C57BL/6 background. These mice were bred with *Cx3cr1^{CreERT}* driver mice (# 021160; Jackson Lab). To induce the Cre recombinase activity, tamoxifen (T5648; Millipore-Sigma) dissolved in corn oil (C8267; Millipore-Sigma), was intraperitoneally injected (1 mg/ml) in 5–6-wk-old mice for five consecutive days. For all mice procedures, age- and sex-matched mice were used for the experiments. All animal works were carried out following the McGill University guidelines directed by the Canadian council of animal care.

### Microglia isolation

To isolate microglia, adult mice were anesthetized by administrating isoflurane and perfused with ice-cold PBS. The brain was isolated, chopped into fragments, and enzymatically dissociated using a neural tissue dissociation kit (Miltenyi Biotec Inc.). To remove myelin debris, cells were spun down in 30% percoll gradient for 20 min at 600*g* (accel 5 and decel 1). The myelin layer was carefully discarded, and pellets were filtered through 70 μm strainer and washed two times with 1× HBSS. Subsequently, cells were stained in FACs buffer with targeted antibodies.

### Flow cytometry

To isolate microglia, cells were first Fc-blocked with anti-CD16/32 (1:200; BD Biosciences). Cells were then stained with anti-CD11b APC (1:200; BioLegend), anti-CD45 BV786 (1:200; BD Biosciences), anti-CD11c PE (1:400; BioLegend), and anti-MHCII BUV737 (1:200; BD Biosciences) for 30 min at 4°C. Cells were washed with PBS and stained with LIVE/DEAD Fixable Aqua Dead Cell Stain (1:500; Invitrogen) for 20 min. CD11b+/CD45inter microglia were sorted using a FACSAria Fusion (BD Bioscience). Cells were washed and stained for LIVE/DEAD Fixable far-red dead Cell Stain (1:500; Invitrogen) for 20 min and washed with PBS. Cells were analyzed on LSR Fortessa and FlowJo software (Tree Star).

### Cuprizone diet

5–6-wk-old *PRMT1^{FL}*, *PRMT1^{Cx3cr1-KO}*, *PRMT5^{FL}*, and *PRMT5^{Cx3cr1-KO}* mice were injected with tamoxifen and 2 wk later the cuprizone diet (CPZ) was initiated. The 0.2% CPZ (Millipore-Sigma) was mixed with the powdered standard diet (Envigo Teklad) and fed for 5 wk. To induce remyelination, mice were fed with standard chow for 1 or 3 wk.

### Immunohistochemical analysis for in vivo brain section

Anesthetized mice were first perfused with PBS followed by 4% PFA. Brains were isolated and incubated in 4% PFA for overnight and

subsequently placed on graded sucrose (10%, 20%, 30%) for 24 h at 4°C each. The brains were embedded with OCT compound and snap-frozen in a mixture of dry ice and isopentane. Next, 12 $\mu$m of serial coronal section of the brain was sliced with a cryostat (Leitz Camera) and mounted immediately onto Superfrost (+) slides (Fisherbrand). Immunofluorescent staining was performed by blocking the sections with 5% BSA containing 0.3% Triton X-100 in PBS for 1 h. The following primary antibodies were incubated overnight at 4°C: Iba-1 (1:200; WAKO), Iba-1 (1:200; Millipore), anti-GFAP (1:300; Abcam), anti-Olig2 (1:200; NovusBiologicals), and anti-Mac2 (1:200; Cedarlane) antibodies. Sections were washed three times with PBS and subsequently incubated for 1 h with the following secondary conjugated antibodies at RT: Alexa Fluor 488 (1: 200; Invitrogen) and Alexa Fluor 568 (1:200; Invitrogen). Counterstain was performed with DAPI for 30 s and mounted. For floating sections, 30 $\mu$m of the coronal section was sliced and immersed in PBS and followed the procedure described above. The presence of myelin was observed using a Black-Gold II staining kit (TR-100-BG; Biosensis) according to the manufacture's instruction (Millipore).

### 3D reconstruction of microglia

Free-floating 30 $\mu$m of the coronal sections of brain were stained with anti-Iba-1 overnight at 4°C, and secondary antibody was incubated for 2 h at RT. Nuclei were stained with DAPI. Confocal images were taken at ×40 oil immersion objective with 1-$\mu$m interval for 20 $\mu$m depth using LSM880 Zeiss microscope. Microglia structure was rendered using IMARIS software.

### TEM analysis

Mice were anesthetized and first perfused with 50 ml of ice-cold PBS followed by 100 ml of fixative solution (2.5% glutaraldehyde, 2.0% PFA in 0.1 M sodium cacodylate buffer at pH 7.4). Brains were isolated and incubated overnight in fixative solution at 4°C. The CC were dissected out and postfixed in 1% aqueous $OsO_4$ (Mecalab) with 1.5% aqueous potassium ferrocyanide for 2 h. UltraCut E ultramicrotome (Reichert-Jung) was used to cut the tissues into 90–100-nm sections and placed in the 200-mesh copper grid (Electron Microscopy Sciences). Myelin layer was captured using an FEI Tecnai 12 120 kV TEM equipped with an AMT XR80C 8 megapixel CCD camera (McGill University, Department of Anatomy and Cell Biology). At least 10 images were taken, and percentage of myelinated axons was quantified.

### Protein extraction and immunoblotting

Cells were lysed with RIPA buffer (50 mM Tris, pH 7.4, 150 mM NaCl, 1 mM DTT, 1% NP40, 0.1% SDS, and 0.5% sodium deoxycholate) and placed on ice for 30 min. Lysed extracts were centrifuged at 20,913$g$ for 10 min in 4°C. Protein concentration was quantified using Rapid Gold BCA Protein Assay Kit (PIA53227, Thermo Fisher Scientific). Equal amounts of proteins were separated using SDS–PAGE and transferred to nitrocellulose membranes using an immunoblot TurboTransfer system (Bio-Rad). Membranes were blocked with 5% skim milk and incubated with primary antibodies overnight at 4°C. The antibodies used are anti-PRMT1 (07-404; Millipore), anti-PRMT5 (07-405; Millipore), anti-H4R3me2a (ab194603; Abcam), anti-

H4R3me2s (ab5823; Abcam), and $\beta$-actin (A3853; Sigma-Aldrich). Appropriate HRP-conjugated secondary antibodies were applied for 1 h at RT. Proteins were visualized by Western Lightning Plus ECL (PerkinElmer).

### RNA-seq library preparation: synthesis of cDNA

RNA in 2X SuperScript III buffer was incubated for 1 min at 50°C on a PCR cycler with 2.5 $\mu$l of the following mix: 1.5 $\mu$g Random Primer (48190-011; Life Technologies), 10 $\mu$M Oligo d(T) (18418020; Life Technologies), 10 units SUPERase-In (AM2696; Ambion), 4 mM dNTP mix (18427088; Life Technologies), in water. Samples were then immediately placed on ice for 5 min. First-strand synthesis was then performed by incubation at 25°C for 10 min and 50°C for 50 min on a PCR cycler with 7.6 $\mu$l of the following mix: 0.2 $\mu$g actinomycin (A1410; Sigma-Aldrich), 13.15 mM DTT (Life Technologies), 0.026% Tween-20, 100 units SuperScript III (kit 18080-044; Life Technologies), in water. After incubation, RNA/DNA complexes were isolated by adding 36 $\mu$l of Agencourt RNAClean XP beads (A63987; Beckman Coulter) and incubated for 10 min at RT and then 10 min on ice. Samples were then placed on a magnet and beads were washed twice with 150 $\mu$l of 75% EtOH. After washings, the beads were air-dried for 10–12 min and eluted with 10 $\mu$l of water. Second-strand synthesis was then performed. RNA/DNA samples in 10 $\mu$l of water was incubated for 2.5 h at 16°C with 5 $\mu$l of the following mix: 3× Blue Buffer (Enzymatics), 1.0 $\mu$l PCR mix (77330; Affymetrix), 2.0 mM dUTP (77206; Affymetrix), 1 unit RNAseH (Y9220L; Enzymatics), 10 units DNA Polymerase I (P7050L; Enzymatics), 0.03% Tween-20, in water. DNA was then purified by addition of 1.5 $\mu$l Sera-Mag SpeedBeads (651520505025; Thermo Fisher Scientific), resuspended in 30 $\mu$l 20% PEG 8000/2.5 M NaCl, incubated at RT for 15 min, and placed on a magnet for two rounds of bead washing with 80% EtOH. Beads were then air-dried for 10–12 min and DNA was eluted from the beads by adding 40 $\mu$l of water. The supernatant was then collected on a magnet and placed on ice or stored at –20°C until DNA blunting, poly(A)-tailing, and adapter ligation (see below).

### RNA-seq and ChIP-seq final library preparation

Sequencing libraries were prepared from recovered DNA (ChIP) or generated cDNA (RNA) by blunting, A-tailing, and adapter ligation as previously described using barcoded adapters (NextFlex; Bioo Scientific) (Gosselin et al, 2014, 2017). Before final PCR amplification, RNA-seq libraries were digested by 30 min of incubation at 37°C with Uracil DNA Glycosylase (final concentration of 0.134 units per $\mu$l of library volume; UDG, Enzymatics G5010L) to generate strand-specific libraries. Libraries were PCR-amplified for 12–15 cycles and size selected for fragments (200–400 bp for ChIP-seq, 200–500 for RNA-seq) by gel extraction (10% TBE gels, Life Technologies EC62752BOX). RNA-seq and ChIP-seq libraries were single-end sequenced on an Illumina HiSeq 4000 (Illumina) according to manufacturer's instruction.

### Chromatin immunoprecipitation

Chromatin immunoprecipitation for histone modification H3K27ac and was performed as follows, using ~500,000 microglia per assay and n = 2 independent biological replicates per conditions. First

microglia were briefly thawed on ice and lysed by incubation in 1 ml of lysis buffer (0.5% IGEPAL CA-630, 10 mM Hepes pH 7.9, 85 mM KCl, 1 mM EDTA, pH 8.0, in water) for 10 min on ice. Lysates were centrifuged at 800 RCF for 5 min at 4°C, and pellets were resuspended in 200 μl of sonication/immunoprecipitation buffer (10 mM Tris–HCl, pH 7.5, 100 mM NaCl, 0.5 mM EGTA, 0.1% deoxycholate, and 0.5% sarkosyl, in water). Sonication was performed with a Bioruptor Standard Sonicator (Diagenode) and consisted of two rounds of 15 min each, alternating stages of 30 s "sonication-on" with 60 s "sonication-off." 22 μl of 10% Triton-X were then added to samples (1% final concentration) on ice, and lysates were cleared by centrifugation for 5 min at 18,000 RCF at 4°C. Two microliters of supernatant were then set aside for input library sequencing controls. Supernatants were then immunoprecipitated on a rotator for 2 h at 4°C with H3K27ac antibody (Active Motif, 39685, 2.5 μg per sample) pre-bound to 17 μl Protein A Dynabeads (10001D; Life Technologies).

Immunoprecipitates were washed three times each on ice with ice-cold wash buffer I (150 mM NaCl, 1% Triton X-100, 0.1% SDS, 2 mM EDTA, pH 8.0 in water), wash buffer III (10 mM Tris–HCl, 250 mM LiCl, 1% IGEPAL CA-630, 0.7% deoxycholate, and 1 mM EDTA in water), and TET (10 mM Tris–HCl, pH 7.5, 1 mM EDTA, pH 8.0, and 0.1% Tween-20, in water) and eluted with 1% SDS/TE at RT in a final volume of 100 μl. Reverse-crosslinking was then performed on immunoprecipitates and saved input aliquots. First, 6.38 μl of 5 M NaCl (final concentration 300 mM) was added to each sample immersed in 100 μl SDS/TE. Crosslinking was then reversed by overnight incubation at 65°C in a hot air oven. Potentially contaminated RNA was then digested for 1 h at 37°C with 0.33 mg/ml RNase A, proteins were digested for 1 h at 55°C with 0.5 mg/ml proteinase K, and DNA was extracted using Sera-Mag SpeedBeads (6515205050250; Thermo Fisher Scientific).

### ChIP-sequencing and bulk RNA-sequencing

Single end reads of length 76 were first trimmed off the adapter sequence as well as the polyA tail and then mapped through the Bowtie2 short read aligner v2.4.1 to the hg19 reference genome from the University of California Santa Cruz (UCSC) (Lander et al, 2001; Langmead & Salzberg, 2012). Quality control checks on the raw sequence data were carried out through the FASTQC software v0.11.9. BAM files were sorted and indexed, and duplicate reads were removed through SAMtools v0.1.19. Peak calling and motif enrichment for ChIP-seq experiments relative to the input was carried out by the Hypergeometric Optimization of Motif Enrichment (HOMER) software v4.11 in histone or super-enhancer mode using default parameters (Heinz et al, 2010). Average read coverage profiles across TSS, TTS or gene body regions and corresponding heatmaps were plotted with the NGS PLOT software (Shen et al, 2014). Density plots of reading coverage were made using the Integrative Genomics Viewer (IGV) software. Gene expression was quantified for RNA-seq experiments using HOMER, and the differential expression relative to the control was computed by DESeq2 (Love et al, 2014). Upregulated or down-regulated genes were defined as having a false discovery rate smaller than 0.05 and absolute log fold change larger than one.

### Single-cell RNA sequencing

Mice brains were harvested at 5 wk of CPZ diet and sorted according to CD11b[+] and CD45[+] by BD FACSAria III (BD Biosciences). The single-cell library was prepared with the McGill Genome center using GemCode Single-Cell Instrument (10× Genomics), and Single Cell 3′ Library & Gel Bead Kit v2. The SPRIselect was used to purify the libraries. Quality was assessed using the size distribution and yield (LabChip GX Perkin Elmer) was quantified by qPCR (KAPA Biosystems Library Quantification Kit for Illumina platforms P/N KK4824). Libraries were sequenced using the Illumina Nova-Seq6000 at IGM Genomics Center (UCSD). Paired end reads of length 101 were aligned to the mm10 mouse genome using Cell Ranger v3.1.0 program (10× Genomics, https://support.10xgenomics.com/single-cell-gene-expression/software/pipelines/latest/what-is-cell-ranger). Afterwards, the Seurat v3.2.0 software was used to carry out quality control as well as for pre-processing analysis (Stuart et al, 2019). Cells containing less than 200 genes and more than 6,000 genes were removed. Cells containing more than 10% of the mitochondria genes were filtered out. Reads counts were then normalized using the "LogNormalize" method. Next, we calculated highly variable features between cells, by applying the linear transformation. The dimensionality of the dataset was determined using an elbow plot. The cluster of the cells were visualized using a Uniform Manifold Approximation and Projection (UMAP). The scVelo v0.2.2 python module was used to perform RNA velocity analysis. Monocle 2 was used to analyze the single-cell pseudotime trajectories (Qiu et al, 2017).

### Quantification and statistical analysis

All experiments were statistically tested using GraphPad Prism 6. All values were represented as means ± SEM. The specific tests used for analysis are indicated in the figure legends. The number of replicates and the animals are described in the figure legends. No blinding was performed during the animal experiments procedure and all the experiments were randomized. No outliers were excluded.

## Data Availability

The RNA-seq data from this publication have been deposited to Gene Expression Omnibus under the accession number GSE201145. The scRNA-seq data from this publication have been deposited to Gene Expression Omnibus under the accession number GSE199574. The ChIP-seq data from this publication have been deposited to Gene Expression Omnibus under the accession number GSE205309.

## Supplementary Information

## Acknowledgements

This work was funded by a Canadian Institute of Health Research FDN-154303 awarded to S Richard. J Lee is a recipient of a Lady Davis Institute/TD Bank Studentship award, Fonds de recherche du Québec Santé (FRQS) and Mogam Foundation Award. We thank S Kelly Sears and Jeannie Mui at the facility for electron microscopy research of McGill university for expert technical assistance in microscope operation and data collection.

### Author Contributions

J Lee: conceptualization, data curation, formal analysis, validation, investigation, visualization, methodology, and writing—original draft, review, and editing.
OD Villarreal: resources, data curation, software, formal analysis, validation, visualization, and methodology.
YC Wang: resources and software.
J Ragoussis: resources, software, and methodology.
S Rivest: conceptualization and methodology.
D Gosselin: conceptualization, resources, data curation, supervision, investigation, methodology, and writing—original draft, review, and editing.
S Richard: conceptualization, resources, data curation, supervision, funding acquisition, project administration, and writing—original draft, review, and editing.

### Conflict of Interest Statement

The authors declare that they have no conflict of interest.

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
