## [Reviewer comments · Life Science Alliance]

Life Science Alliance

PRMT1 is required for the generation of MHC-associated microglia and remyelination in the CNS

Stéphane Richard, Jeesan Lee, Oscar Villarreal, Yu Chang Wang, Jiannis Ragoussis, Serge Rivest, and David Gosselin
DOI: <https://doi.org/10.26508/lsa.202201467>

Corresponding author(s): *Stéphane Richard, McGill University*

Review Timeline:	Submission Date:	2022-03-29
	Editorial Decision:	2022-05-06
	Revision Received:	2022-05-22
	Editorial Decision:	2022-05-27
	Revision Received:	2022-06-03
	Accepted:	2022-06-03

Scientific Editor: Novella Guidi

Transaction Report:

May 6, 2022

Re: Life Science Alliance manuscript #LSA-2022-01467

Prof. Stephane Richard
McGill University
Departments of Oncology and Medicine
Lady Davis Institute for Medical Research
3755 Cote-Ste-Catherine Road
Montreal, Quebec H3T 1E2
Canada

Dear Dr. Richard,

Thank you for submitting your manuscript entitled "PRMT1 is required for the generation of MHC-associated microglia and remyelination in the CNS" to Life Science Alliance. The manuscript was assessed by expert reviewers, whose comments are appended to this letter. We invite you to submit a revised manuscript addressing the Reviewer comments.

Thank you for this interesting contribution to Life Science Alliance. We are looking forward to receiving your revised manuscript.

Sincerely,

B. MANUSCRIPT ORGANIZATION AND FORMATTING:

Reviewer #1 (Comments to the Authors (Required)):

Summary:

Myelin basic protein was one of the first arginine methylated proteins to be identified. It is believed that MBP methylation plays a role in myelination. In this study, the groups led by David Gosselin and Stephane Richard investigated the role of two key PRMTs in remyelination. PRMT1 and PRMT5 account for over 80% of the ADMA and SDMA deposited in the cell, respectively. They generate microglia-specific knockout of PRMT1 and PRMT5 and used a cuprizone (CPZ) induced de- and re-myelination mouse model to investigate the roles of these two PRMTs in this process - only PRMT1 conditional knockout showed a phenotype. They also performed single-cell RNA seq, which revealed major changes in microglia composition after PRMT1 knockout, particularly with a 24% gain in the E population. They further show that MHCII expression is increased after CPZ-treatment, and that PRMT1 is required for the induced expression.

Critique:

This is an in-depth in vivo study of the roles of PRMT1 and PRMT5 in remyelination. In the context of microglia, there is a clear role for PRMT1 but not PRMT5 in this process. In addition, they show that PRMT1 is a positive regulator of MHCII expression in microglia. Both these observations are novel. The manuscript is clearly written, and easy to follow. However, there are a number of issues that need to be addressed before this manuscript is suitable for publication.

The following issues need to be addressed:

1. What is the cause of the dramatic remyelination defect seen in PRMT1 KO mice after CPZ-treatment (Figure 1 E)? Is it due to the loss of MBP methylation?
2. In Nature Comm (2018), Guccione and Casaccia reported that PRMT5 regulates myelination, which was not observed in this study. This paper should be discussed and referenced.
3. In Scientific Reports (2016), Zhiwen Fen reported that PRMT1 is a repressor of MHCII transcription. They show that this is due to the methylation of CIITA by PRMT1. These published results seem to contradict the finding reported here that show PRMT1 is needed for the induction of MHCII expression. This discrepancy may be due to the fact that one study used macrophages and the other microglia. This paper should be discussed and referenced.
4. On page 11, they state that "...indicating that histone methylation by PRMT1 is required for the subsequent deposition of H3K27ac at specific promoters...". This is an over interpretation of their data. It is not clear that histone arginine methylation is important for H3K27ac. It could be the arginine methylation, by PRMT1, of transcription factors or coactivators (like the HATs themselves), that results in increased H3K27ac.

Reviewer #3 (Comments to the Authors (Required)):

Here, by using scRNAseq and animal disease models with cell-type specific gene deletion, Lee and colleagues describe that the appearance of a specific microglia subpopulation requires PRMT1, but not PRMT5. In general, the data is convincing, but I would suggest several points to be considered to add further value to this article before publication.

Major points

1. How does PRMT1 deficiency affect microglial phenotype under homeostatic condition? Although the authors mentioned in the discussion that they didn't see any significant changes in microglia between wild-type and PRMT1Cx3cr1-KO mice, there are no data shown in the manuscript to support this notion. In fact, the expression level of MHCII seems to be different between PRMT1FL and PRMT1Cx3cr1-KO under normal diet (Figures 4A and B). The related data should be implemented.
2. Similarly, the data for Igf1 expression (e.g. violin plot) is missing.
3. The recent paper (PMID: 35260865) has shown that enzymatic treatment during cell isolation has a strong impact on gene expression in microglia. The authors should mention and discuss this point.

4. In Figure 2, how many microglia isolated from how many mice per genotype are displayed?

Minor

1. Different drugs (tamoxifen, 4-hydroxytamoxifen) for pulsing were described in the manuscript (method vs text), which should be clarified.
2. Page 4; In the "Introduction", the condition of mouse line (e.g. PRMT1FL/FL;Cx3cr1-KO) should be explained at first appearance.
3. Page 5: PRMT1FL/FL;Cx3cr1-KO (PRMT1Cx3cr1-KO) would be PRMT1FL/FL;Cx3cr1-CreERT (PRMT1Cx3cr1-KO)? The same is for PRMT5 FL/FL;Cx3cr1-KO (PRMT5FL/FL;Cx3cr1-KO).

Reviewer #1 (Comments to the Authors (Required)):

Summary:

Myelin basic protein was one of the first arginine methylated proteins to be identified. It is believed that MBP methylation plays a role in myelination. In this study, the groups led by David Gosselin and Stephane Richard investigated the role of two key PRMTs in remyelination. PRMT1 and PRMT5 account for over 80% of the ADMA and SDMA deposited in the cell, respectively. They generate microglia-specific knockout of PRMT1 and PRMT5 and used a cuprizone (CPZ) induced de- and re-myelination mouse model to investigate the roles of these two PRMTs in this process - only PRMT1 conditional knockout showed a phenotype. They also performed single-cell RNA seq, which revealed major changes in microglia composition after PRMT1 knockout, particularly with a 24% gain in the E population. They further show that MHCII expression is increased after CPZ-treatment, and that PRMT1 is required for the induced expression.

Critique:

This is an in-depth in vivo study of the roles of PRMT1 and PRMT5 in remyelination. In the context of microglia, there is a clear role for PRMT1 but not PRMT5 in this process. In addition, they show that PRMT1 is a positive regulator of MHCII expression in microglia. Both these observations are novel. The manuscript is clearly written, and easy to follow. However, there are a number of issues that need to be addressed before this manuscript is suitable for publication.

The following issues need to be addressed:

1. What is the cause of the dramatic remyelination defect seen in PRMT1 KO mice after CPZ-treatment (Figure 1 E)? Is it due to the loss of MBP methylation?

REPLY: We do not know the exact cause of the dramatic remyelination defect in PRMT1^{Cx3cr1-KO} mice. We suspect it may be a defect in clearing myelin debris, or the persistent microgliosis observed during the remyelination period which hinders the recruitment of OPCs and their survival. Although MBP methylation at R107 by PRMT5 is important it is likely not at play in our mouse models.

We have added a reference to cite that MBP is methylated in OPCs.

“Furthermore, it is also required in oligodendrocyte progenitor cells for their differentiation and survival for CNS myelination by epigenetically regulating genes required for differentiation (Scaglione et al 2018) and myelin basic protein (Branscombe et al 2001).”

2. In Nature Comm (2018), Guccione and Casaccia reported that PRMT5 regulates myelination, which was not observed in this study. This paper should be discussed and referenced.

REPLY: This was an oversight on our part. We now cite the paper.

“Furthermore, it is also required in oligodendrocyte progenitor cells for their differentiation and survival for CNS myelination by epigenetically regulating genes required for differentiation (Scaglione et al 2018) and myelin basic protein (Branscombe et al 2001).”

3. In Scientific Reports (2016), Zhiwen Fen reported that PRMT1 is a repressor of MHCII transcription. They show that this is due to the methylation of CIITA by PRMT1. These published results seem to contradict the finding reported here that show PRMT1 is needed for the

induction of MHCII expression. This discrepancy may be due to the fact that one study used macrophages and the other microglia. This paper should be discussed and referenced.

REPLY: we have added the following text to discuss.

“Although, we propose a positive role for PRMT1 in MHC II gene expression in microglia, an opposite role for PRMT1, namely as a repressor of MHC II transcription, has been reported in macrophages by promoting the arginine methylation and the degradation of the class II transactivator (CIITA) (Fan et al 2017). In macrophages PRMT1 is needed for their differentiation into a more anti-inflammatory phenotype via H4R3me2a methylation at the PPARγ promoter (Tikhanovich et al 2017). Moreover, methylation of TBK1 by PRMT1 is needed for its oligomerization and stimulation of type I interferon production (Yan et al 2021). As a result of this, myeloid-specific PRMT1 knockout mice are more susceptible to viral infection (Tikhanovich et al 2017, Yan et al 2021).”

4. On page 11, they state that "...indicating that histone methylation by PRMT1 is required for the subsequent deposition of H3K27ac at specific promoters...". This is an over interpretation of their data. It is not clear that histone arginine methylation is important for H3K27ac. It could be the arginine methylation, by PRMT1, of transcription factors or coactivators (like the HATs themselves), that results in increased H3K27ac.

REPLY: Indeed, it was an over interpretation. The sentence was rewritten.

“These findings suggest PRMT1 is required for the subsequent deposition of H3K27ac at specific promoters (Figure 5C), either by the deposition of H4R3me2a or the methylation of transcription factors or coactivators (e.g. histone acetyltransferases) that results in H3K27ac.”

Reviewer #3 (Comments to the Authors (Required)):

Here, by using scRNAseq and animal disease models with cell-type specific gene deletion, Lee and colleagues describe that the appearance of a specific microglia subpopulation requires PRMT1, but not PRMT5. In general, the data is convincing, but I would suggest several points to be considered to add further value to this article before publication.

Major points

1. How does PRMT1 deficiency affect microglial phenotype under homeostatic condition? Although the authors mentioned in the discussion that they didn't see any significant changes in microglia between wild-type and PRMT1^{Cx3cr1-KO} mice, there are no data shown in the manuscript to support this notion. In fact, the expression level of MHCII seems to be different between PRMT1^{FL} and PRMT1^{Cx3cr1-KO} under normal diet (Figures 4A and B). The related data should be implemented.

REPLY: Indeed, the reviewer is correct and we have added the following text to address the issue.

“We did not observe any significant change in the gene expression between microglia from wild type and PRMT1^{Cx3cr1-KO} mice, however, we did observe a slight decrease in MHC II expression in the PRMT1-deficient microglia on a normal diet (Figure 4A, 4B), suggesting PRMT1 may affect the homeostatic function of microglia.”

2. Similarly, the data for Igf1 expression (e.g. violin plot) is missing.

REPLY: We now present the data in Supplementary Figure S4 and the expression of Igf1 does not decrease with the absence of CD11cHi in *PRMT1^{Cx3cr1-KO}* mice. The following was added in the discussion.

*“Analysis of our scRNA-seq data revealed that the MHC-associated cluster arose with de/remyelination pathology and exclusively expressed CD11c. Igf1 was expressed in all clusters, except cluster D, with highest expression being in clusters A1, A2, B1 and B2 that do not change much in *PRMT1^{Cx3cr1-KO}* mice (Supplemental Figure S4A). Although a loss of CD11c+ microglia population, as defined by protein expression, in *PRMT1^{Cx3cr1-KO}* microglia was observed, no significant decrease Igf1 mRNA expression level was noted that could account for the remyelination phenotype we observe. Igf1 expression increased significantly with CPZ diet in both WT_CD11cLo and KO_CD11cLo. The increase in Igf1 mRNA of the KO_CD11cLo was similar to the WT_CD11cHi microglia (Supplemental Figure S4B). Therefore, absence of *PRMT1* does not compromise production of Igf1 by microglia which suggests that alternative defective mechanisms limit remyelination.”*

3. The recent paper (PMID: 35260865) has shown that enzymatic treatment during cell isolation has a strong impact on gene expression in microglia. The authors should mention and discuss this point.

REPLY: We now cite Marsh et al., 2022 and added the following paragraph in the discussion to address cell isolation influence on gene expression.

“We did not add a transcriptional inhibitor (e.g., Flavoperidol or Actinomycin D) to our buffers during the microglial isolation procedure. We did not find evidence that expression of CCL3, CCL4, JUN, and HSPA1A mRNAs, which are markers of spurious transcriptional induction associated with enzymatic-based isolation of microglia (Marsh et al 2022) drove a specific cluster based on single-cell RNA-seq data. For example, CCL3 and CCL4 were elevated in clusters B2 and C2, JUN in D, and HSPA1A in F (Supplemental Table S1). Given this, genotypes and experimental conditions, but not the isolation procedure used, were likely the main variables driving the clusters identified in our analyses.”

4. In Figure 2, how many microglia isolated from how many mice per genotype are displayed?

REPLY: We have added following text to the figure2 legend

*“sc-RNA seq was performed from *PRMT1^{FL}* (n=2) and *PRMT1^{Cx3cr1-KO}* (n=2) mice during the CPZ diet 5 weeks. Representative UMAP visualization of 6,000 microglia from each of the genotypes (n=1) showing 9 distinct clusters of microglia.”*

Minor

1. Different drugs (tamoxifen, 4-hydroxytamoxifen) for pulsing were described in the manuscript (method vs text), which should be clarified.

REPLY: We used tamoxifen (Sigma; T5648) and corrected this in the manuscript.

2. Page 4; In the "Introduction", the condition of mouse line (e.g. *PRMT1^{FL/FL};Cx3cr1-KO*) should be explained at first appearance.

REPLY: We explained at first appearance.

“Mice with microglia deficient for PRMT1 using the Cx3cr1^{CreERT} driver were generated (PRMT1^{FL/FL;Cx3cr1-CreERT}). Lack of the MHC-associated microglia cluster in tamoxifen (TAM) treated PRMT1^{FL/FL;Cx3cr1-CreERT} mice correlates with a failure to induce CNS remyelination after demyelination induced by CPZ.”

3. Page 5: PRMT1FL/FL;Cx3cr1-KO (PRMT1Cx3cr1-KO) would be PRMT1FL/FL;Cx3cr1-CreERT (PRMT1Cx3cr1-KO)? The same is for PRMT5 FL/FL;Cx3cr1-KO (PRMT5FL/FL;Cx3cr1-KO).

REPLY: We have corrected the text as follows: *“Microglia isolated from tamoxifen (TAM) injected PRMT1^{FL/FL;Cx3cr1-CreERT} (PRMT1^{Cx3cr1-KO}) and PRMT5^{FL/FL;Cx3cr1-CreERT} (PRMT5^{Cx3cr1-KO})... ”*

May 27, 2022

RE: Life Science Alliance Manuscript #LSA-2022-01467R

Prof. Stephane Richard
McGill University
Departments of Oncology and Medicine
Lady Davis Institute for Medical Research
3755 Cote-Ste-Catherine Road
Montreal, Quebec H3T 1E2
Canada

Dear Dr. Richard,

Thank you for submitting your revised manuscript entitled "PRMT1 is required for the generation of MHC-associated microglia and remyelination in the CNS". We would be happy to publish your paper in Life Science Alliance pending final revisions necessary to meet our formatting guidelines.

- please correct the typo on page 14 indicated by Reviewer 3
- please upload your main and supplementary figures as single files
- please add the Twitter handle of your host institute/organization as well as your own or/and one of the authors in our system
- please make sure that all authors are added in our system and that the author names in the manuscript and our system match
- please add a separate Data availability section providing accession number for the deposited RNAseq and CHIP-seq data

Figure Check:

- Please expand Figure S3 legend and remove the letter A from the panel figure since it is the only one

A. FINAL FILES:

B. MANUSCRIPT ORGANIZATION AND FORMATTING:

Sincerely,

Reviewer #1 (Comments to the Authors (Required)):

All my concerns were addressed by the authors.

Reviewer #3 (Comments to the Authors (Required)):

The authors have sufficiently revised the manuscript, and I have no further comment on this, except a subtle typo "expect" in page 14.

June 3, 2022

RE: Life Science Alliance Manuscript #LSA-2022-01467RR

Prof. Stéphane Richard
McGill University
Departments of Oncology and Medicine
Lady Davis Institute for Medical Research
3755 Cote-Ste-Catherine Road
Montreal, Quebec H3T 1E2
Canada

Dear Dr. Richard,

Thank you for submitting your Resource entitled "PRMT1 is required for the generation of MHC-associated microglia and remyelination in the CNS". It is a pleasure to let you know that your manuscript is now accepted for publication in Life Science Alliance. Congratulations on this interesting work.

DISTRIBUTION OF MATERIALS:

Again, congratulations on a very nice paper. I hope you found the review process to be constructive and are pleased with how the manuscript was handled editorially. We look forward to future exciting submissions from your lab.

Sincerely,
